# Determination of Nitrate Migration and Distribution through Eutric Cambisols in an Area without Anthropogenic Sources of Nitrate (Velika Gorica Well Field, Croatia)

**Patricia Buškulić** [iD], **Jelena Parlov** *[iD], **Zoran Kovač** [iD], **Tomislav Brenko** [iD] and **Marija Pejić** [iD]

Faculty of Mining, Geology and Petroleum Engineering, University of Zagreb, 10000 Zagreb, Croatia; patricia.buskulic@rgn.unizg.hr (P.B.); zoran.kovac@rgn.unizg.hr (Z.K.); tomislav.brenko@rgn.unizg.hr (T.B.); marija.pejic@rgn.unizg.hr (M.P.)
* Correspondence: jelena.parlov@rgn.unizg.hr

**Abstract:** Natural potential sources of nitrate contamination involve decaying of organic matter, bacterial production, atmospheric deposition, and soil N. The study presents the first results of nitrate distribution and migration through soil horizons of the Eutric Cambisols, one of the most common soils developed in the area of the Zagreb aquifer and situated in an area without potential anthropogenic sources of nitrate (first sanitary protection zone of the Velika Gorica well field). A total of 16 parameters of soil water and 16 parameters of soil were used to conduct statistical techniques and analyse associated factors within the soil zone. The results indicate that in the deepest soil horizon, nitrogen is present mostly as nitrate due to nitrification under aerobic conditions which promote stability and the potential for nitrate transport. It was found that nitrate concentrations are the result of soil N nitrification, caused by a $NO_3^-/Cl^-$ molar ratio higher than 1 and the absence of precipitation isotopic signature. The results also show that within the coarse-grained Eutric Cambisols N primarily migrates to deeper parts of unsaturated zone in the form of nitrate and nitrite.

**Keywords:** nitrogen cycle; soil zone; Eutric Cambisols; Zagreb aquifer

## 1. Introduction

Nitrate ion is a main form of nitrogen (N) and one of the more common contaminants in natural environments [1,2]. Although some plants can directly use atmospheric N, ammonium ($NH_4^+$) and nitrate ($NO_3^-$) are inorganic forms of N that are usable to most plants [3]. Excluding septic systems, animal waste and commercial fertilizer, significant natural potential sources of $NO_3^-$ contamination involve decaying of organic matter (OM), bacterial production, atmospheric deposition, and soil N [4,5]. Tracing the sources and transformations of $NO_3^-$ is crucial for gaining insights into water quality protection and better understanding of the N cycling [2]. Furthermore, it has been shown that hydrochemistry (e.g., $NO_3^-$, $Cl^-$) and isotopes (e.g., $^{18}O$-$H_2O$) can provide important information for differentiating between $NO_3^-$ sources and the processes involved in N cycling [2]. The deuterium excess (d-excess) is also a valuable tool for discerning the influences of evaporation and/or mineral dissolution trends [5]. Additionally, it is an important tool in assessing the mean residence time of soil water and recharge processes [6].

$NO_3^-$ is considered the most oxidized, stable, and mobile form of N species in solution [7]. The accumulation of $NH_4^+$ in soils is not common, as it undergoes rapid conversion by soil microbes [3]. The presence of $NH_4^+$ at some depth in the unsaturated zone indicated that reducing conditions might be present [8]. Due to oversaturation in the soil, N species are lost to groundwater through leaching, which contributes to groundwater contamination [9]. Soil conditions that enhance the retention of $NH_4^+$ and $NO_3^-$ ions, i.e., a zeolite with a high exchange capacity, offers a solution by absorbing ammonium and slowing down the nitrate leaching [10]. $Cl^-$ tends to behave in a more stable manner,

with the minimum amount of chemical reactions or transformations [11–13] and minimum interactions with subsoil [5] because it is inert to physical, chemical, and microbiological processes [14]. The molar ratio of $NO_3^-/Cl^-$ has been deemed to be a valuable tool for exploring N dynamics and sources [12,13]. Additionally, higher $NO_3^-/Cl^-$ molar ratios suggest that some potential $NO_3^-$ input might have been ascribed to precipitation, fertilizer application, and nitrification of soil N [2].

Soil nitrification is a two-step process performed by living soil microorganism [3,9,15–17]. Ammonia-oxidation (conversion of $NH_4^+$ to $NO_2^-$, i.e., nitritation) is carried out by ammonia oxidizers [18,19], which is widely distributed in most agricultural soils and represents the major contributor to nitrification [20]. The second step is nitrite-oxidation (transformation of $NO_2^-$ into $NO_3^-$, i.e., nitratation) [3], performed by nitrite oxidizers [21,22]. When the rate of nitritation is faster than nitratation, $NO_2^-$ accumulates. When nitratation takes place more rapidly, only a small amount of $NO_2^-$ is produced [23,24]. The presence of $NH_4^+$ and $NO_3^-$ or a significant amount of $NH_4^+$ at some depth in the unsaturated zone is evidence of incomplete nitrification [8]. Conversely, denitrification involves reduction of $NO_3^-$ through the conversion of $NO_3^-$ into $N_2$, $N_2O$, or $NO$, generally under anaerobic conditions [25–27].

Previous studies have shown that the nitrification process in soils depends on many factors, such as soil moisture, temperature, soil pH, organic carbon content, the presence of major oxides and heavy metals, as well as soil texture. Excess water in soils can lead to oxygen limitation, which reduces the rate of nitrification [28], while microbial activity generally increases with increasing temperature [3,9]. The highest denitrification occurs when a combination of high soil moisture and high soil temperature is present, whereas the low soil moisture appears to restrain the degree of denitrification [29]. Soil moisture closes pore spaces, which in turn impairs aeration and reduces the oxygen level. As nitrification is a biochemical oxidation process, low oxygen levels in the soil negatively affect the process of nitrification [9]. During summer months, the assimilation of $NO_3^-$ by plants and denitrification process reduces $NO_3^-$ concentrations [11]. On the other hand, the lowest denitrification occurs when the rainfall abruptly increases, causing enhancement of leaching. Bacterial diversity and community structure are significantly influenced by the pH of the soil [3,30,31]. In acidified soils, the intensity of nitrification is lower compared to soils with higher pH values [32–34]. The optimum activity of ammonia oxidizers and nitrite oxidizers occurs at pH 7.5 and 7.0 [9]. In general, pH values in the topsoil tends to be lower, primarily because the topsoil contains a higher concentration of OM, and the decomposition of OM lowers pH [35]. Organic carbon is another significant factor that influences the rate of nitrification in the soil [32,36,37]. The presence of organic carbon inhibits nitrification by reducing the abundance of ammonia oxidizers, whereas low organic carbon levels in soil enhance the nitrification rate, resulting in higher $NO_3^-$ concentrations [20,38]. Elevated levels of soil organic carbon are associated with increased OM content, which in turn improves permeability and water availability [39]. On the other hand, reduced input of OM along with soil depth tends to decrease total organic carbon [40]. Additionally, other compounds in soil, such as titanium dioxide ($TiO_2$), can reduce the abundance of ammonia-oxidizers and nitrite-oxidizers [41]. The impact of iron (Fe) minerals should also not be ignored, especially oxides, whose influence on soil N transformation processes varies according to soil pH. In the low-pH soil, Fe oxide frequently stimulates nitrification activity, while in the high-pH soil, Fe oxide significantly decreases nitrification rate [42]. Anaerobic $NH_4^+$ oxidation can be linked to ferric iron reduction, resulting in the production of $N_2$, $NO_2^-$ [43], or $NO_3^-$ [44] as the end product. These reactions involve the use of ferric iron ($Fe^{3+}$) as an electron acceptor. Moreover, reactions of manganese (Mn) oxides in soil are similar to Fe and play significant roles in N cycling process [45]. Under oxic conditions, Mn has a toxic effect on microorganisms, whereas under oxygen-depleted conditions, Mn serves as an alternative electron acceptor. Furthermore, toxic elements such as heavy metals often negatively affect nitrification rate in soils [46,47]. Chromium (Cr) increases $NH_4^+$ content and decreases the accumulation of $NO_3^-$ [48]. Nickel (Ni) [49], zinc (Zn) [50], lead

(Pb) [46], arsenic (As) [51], cobalt (Co) [52] and mercury (Hg) [53] have toxic effects on microorganisms and inhibit nitrification processes in soil, i.e., $NH_4^+$ oxidation to $NO_2^-$, leading to a reduction in $NO_3^-$ concentration. Soil texture, which characterizes the size distribution of soil and mineral particles, is a significant factor that affects the accumulation of soil OM [54]. Clay and silt particles are small in size, however they have large specific surface areas and the ability to absorb and protect soil OM by providing stability against microbial mineralization [55,56]. Considering that levels of OM are associated with levels of soil organic carbon, the soils with higher silt and clay content tend to have higher soil organic carbon [56].

On the other hand, nitrification process (i.e., accumulation of $NO_3^-$ ion) can lower soil pH by causing the leaching of $Ca^{2+}$ and $Mg^{2+}$ and reducing their concentrations [57–59]. Conversely, $NH_4^+$ acidifies the soil by directly exchanging base cations [60]. In an oxygen-deficient environment of soil, both nitrification and denitrification processes become more pronounced, which leads to the formation and accumulation of $NO_2^-$ as an intermediate product [61]. In conditions of relatively low soil moisture, the oxygen content is higher, leading to stronger nitrification [61].

The scientific research polygon of the Faculty of Mining, Geology, and Petroleum Engineering, University of Zagreb [62], is located within the first sanitary protection zone of the Velika Gorica well field, situated in the southern part of the Zagreb aquifer, which presents strategic water reserves and the main source of potable water in the Zagreb area protected by the Republic of Croatia. In recent history, previous investigations were focused on various aspects related to $NO_3^-$ contamination in the groundwater of the Zagreb aquifer [63–65]. Despite research conducted in the selected area, there has been no specific focus on investigating the distribution of nitrates through Eutric Cambisols and related geochemical processes. The primary focus of research within the soil and the unsaturated part of the Zagreb aquifer has been on establishing the relationship between permeability and physicochemical properties [66], determining the sorption characteristics of potentially toxic metals [67], evaluating soil water origin [68], and conducting soil water monitoring of multiple soil horizons in Eutric Cambisols at the Velika Gorica site [69].

The objectives of this study have been to evaluate the $NO_3^-$ distribution and migration through pedological profile situated in an area without anthropogenic sources of nitrate. This has been tested by conducting statistical techniques to determine the significant variability within different soil horizon groups and by analysing and characterizing the factors influencing $NO_3^-$ concentration within the soil zone. For this purpose, 16 different parameters of soil water and 16 different parameters of soil have been chosen. One of the specific goals of this research is the definition of a dominant natural N form that infiltrates into the aquifer. The preliminary findings from this research offer a new insight into geochemical processes related to N species transformation occurring within the Eutric Cambisols, which present one of the most common soils developed in the area of the Zagreb aquifer. Long-term goals also involve modelling the flow and transport of N compounds within the soil and unsaturated zone. It is expected that by achieving these goals new measures necessary for effective and sustainable management of the Zagreb aquifer will be adopted.

## 2. Materials and Methods

### 2.1. Site Description

According to Bogunović et al. [70] the research polygon is located in Eutric Cambisols on Holocene deposits (Figure 1). The unsaturated zone thickness at the study site usually ranges from 5 to 8 m and depends on the groundwater levels. At the top of the unsaturated zone, the following soil horizons were identified according to the World Reference Base classification: A (0–0.15 m), 2B (0.15–0.55 m), 3BC (0.55–0.9 m), and C (0.9–1.17 m).

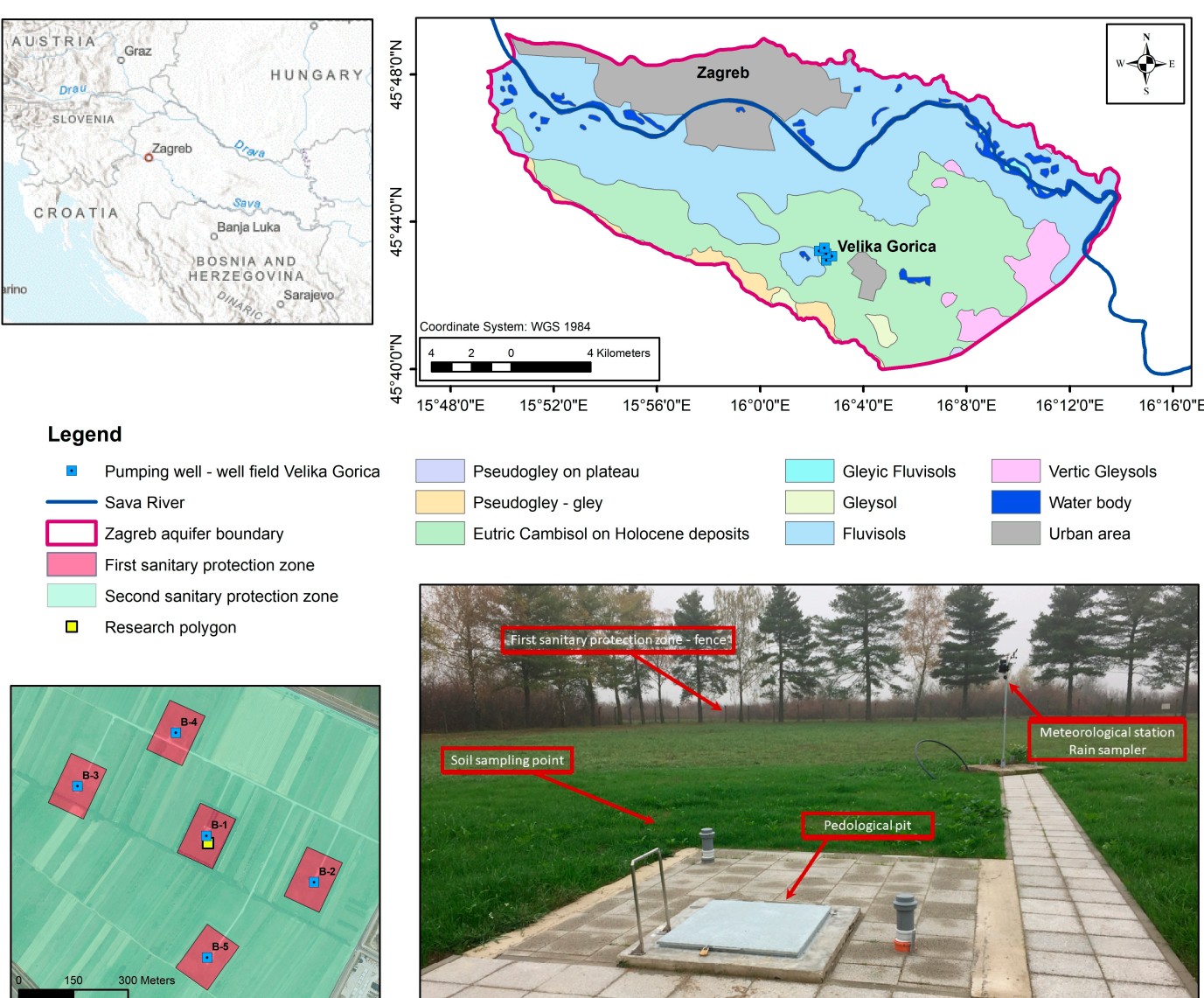

**Figure 1.** Location of the scientific research polygon.

In Table 1, particle size analysis and OM content are shown. OM values are determined for the first three soil horizons, where fine-grained particles are predominant. In accordance with Ružičić et al. [66], the upper 90 cm of the observed profile is predominantly composed of silty and sandy materials, with intermittent clay layers, while gravels dominate throughout the profile until reaching the water table [71]. It is important to highlight that most silt and clay particles, along with the smallest proportion of sand, can be found in the 2B horizon, while the C horizon contains a significant amount of gravel and silt. It can be seen that the A horizon has the highest OM values, while the 2B horizon has the lowest.

**Table 1.** Selected characteristics of the investigated soil profile.

| Soil Horizon | Depth (m) | Gravel (%) | Sand (%) | Silt (%) | Clay (%) | OM (%) |
|---|---|---|---|---|---|---|
| A | 0–0.15 | - | 18.30 | 54.83 | 26.87 | 5.35 |
| 2B | 0.15–0.55 | - | 5.14 | 55.33 | 39.53 | 2.07 |
| 3BC | 0.55–0.9 | - | 13.72 | 50.75 | 35.53 | 3.32 |
| C | 0.9–1.17 | 52.20 | 1.50 | 45.65 | 0.64 | - |

Source: Data from Refs. [66,67,69].

The climate is humid continental, with mean annual precipitation and temperature for the period 2001 to 2020 of about 967 mm and 11.9 °C, respectively, and with monthly average precipitation of around 80 mm [65].

Within the polygon, there is a weather station, rain sampler, and a pedological pit (Figure 1) equipped with various measuring instruments to observe and measure parameters in each soil horizon, as well as the unsaturated and saturated zones.

### 2.2. Data Collection and Sampling

Precipitation and air temperature data at hourly intervals were collected using a meteorological station (Vantage Pro2, Davis Instruments) positioned near the pedological pit. A total of 12 precipitation samples were collected using a Palmex Rain Sampler RS1 (Zagreb, Croatia) [72] in order to determine water stable isotopes and chemical compositions. Four TRIME-PICO 64 probes (IMKO Micromodultechnik GmbH, Ettlingen, Germany) are installed in soil horizons and employed to measure moisture and temperature in soil horizons. Hourly measurements of soil moisture and soil temperature were collected using dataTaker DT80 and globeLog (IMKO Micromodultechnik GmbH) loggers. Soil water samples were sampled from four soil horizons using soil water samplers (suction cups; Eijkelkamp Soil & Water, Giesbeek, The Netherlands) and an integrated automatic vacuum pump unit AVP–100 (UGT GmbH, Müncheberg, Germany) to determine isotopic and chemical compositions. Suction cups are situated at following depths: $-0.08$ (SC1), $-0.33$ (SC2), $-0.75$ (SC3), and $-1.05$ m (SC4). During certain months (July, August, and October), it was not possible to obtain soil water samples from soil water samplers SC3 and SC4, while during the most dry month, i.e., September, not a single soil water sample could be taken. This arose primarily due to low soil water content. For this reason, a total of 38 soil water samples were collected. Additionally, due to very small volume amounts available from the deepest soil water sampler SC4, the chemical composition was analyzed for 35 soil water samples. Samples and data were collected from March 2021 to February 2022. Each soil water and precipitation sample was filtered using a 0.22 μm nylon membrane filter and then moved into high-density polyethylene (HDPE) bottle.

A total of 12 soil samples were collected up to a depth of 1.2 m using Eijkelkamp auger set for soils. The soil profile was excavated near the pedological pit. Soil samples were collected at 10 cm depth intervals in order to measure the following soil parameters: pH, electric conductivity (EC), total organic carbon (TOC), $TiO_2$, $Fe_2O_3$, MnO, heavy metals, and soil texture. The samples were stored in separate plastic bags, transported to the laboratory and air-dried.

### 2.3. Laboratory Measurements

All laboratory measurements were performed at the Laboratory for spectroscopy of the Faculty of Mining, Geology and Petroleum Engineering, University of Zagreb. The concentrations of major anions and cations were determined using a Dionex ion chromatograph (IC). The water stable isotopes ($\delta^2$H-$H_2O$ and $\delta^{18}$O-$H_2O$) were analysed using a Los Gatos Research laser (LWIA-45-EP, San Jose, CA, USA) by laser absorption spectroscopy. The analytical precision was 0.9 ‰ for $\delta^2$H and 0.19 ‰ for $\delta^{18}$O. Values are expressed in permil notation relative to Vienna Standard Mean Ocean Water (VSMOW). The data were analysed and interpreted using the Laboratory Information Management System (LIMS) for Lasers 2015 [73]. D-excess is calculated as d-excess = $\delta^2$H $- 8*\delta^{18}$O [74].

A portion of each soil sample interval was sieved through a 2 mm sieve and homogenized in an agate grinding set. Soil pH in 1M KCl was measured using a pH meter in a 1:5 suspension of soil volume and 1M KCl solution according to ISO10390:2005. EC was measured in a suspension of 1:5 soil volume and $H_2O$. Total carbon (TC) and total inorganic carbon (TIC) were measured with Elementary analyser multi-EA 4000 (Analytik Jena AG, Jena, Germany). TOC was obtained by subtracting TIC from TC. Soil texture was characterized with laser diffraction method using Malvern Mastersizer 3000. The 2000-63-2-μm system was used to determine particle size fractions. Soil particle size classification was

done according to the IUSS Working Group WRB [75]. Geochemical contents of major oxides and microelements were determined using Hitachi XMET 8000 Expert Geo portable X-ray fluorescence (pXRF) instrument. Soil and MiningLE (light elements) calibrations were used. The accuracy of the analysis was controlled by analysing the standard material for soil samples (NIST 2711) in the studied sample batches. Based on five measurements and the use of blanks and standards, the instrumental precision was ±5% or less.

### 2.4. Statistical Analysis

Statistical analyses were performed with the TIBCO software Inc. Statistica (Version 13.5.0.17). Depending on the parametric or non-parametric nature of the data, different statistical analyses are chosen. One-way ANOVA is used for parametric sample data analysis and Kruskal–Wallis (KW) test is used for non-parametric data. Sixteen soil water variables were tested and the main goal of conducting ANOVA or KW test was to establish significant differences within different sampling depths for each variable.

When performing a one-way ANOVA parametric test, there are assumptions that need to be met: dependent variable should follow a normal distribution and the variance should be constant across groups [76–78]. Shapiro–Wilk test is therefore used to check if variable comes from a normal distribution. Levene's test is utilized to test equality of variances in a dataset, i.e., to test the null hypothesis that the samples come from a population with the same variance. The KW test is used when the assumptions of one-way ANOVA are not met [79,80].

The ANOVA is a statistical technique used to assess the variability and determine the variation of the means of a group of data or variables to evaluate statistical significance [77,81]. The KW test is a non-parametric method for testing whether samples are originated from the same distribution [82].

If the ANOVA or KW test yields a statistically significant difference, the post hoc tests, namely, the Tukey Honest Significant Difference test (HSD) and Mann–Whitney U test, are employed to compare parameters between the groups. Tukey HSD test is used after one-way ANOVA test to show comparisons between each pair of groups at a significant level of 0.05 [83,84]. The Mann–Whitney U test is utilized after the KW test and it is comparable with the post hoc Tukey HSD test. The Mann–Whitney U test is used to compare the distribution among different groups of soil sample data.

## 3. Results

### 3.1. Precipitation and Air Temperature

Monthly values of precipitation and air temperature (Figure 2a), as well as precipitation chemical composition (Figure 2b) are shown. Monthly precipitation varied from 29.8 to 102 mm, and the driest months were June, August, September, January, and February. Mean air temperature ranges from 1.16 to 23.27 °C. Monthly analyses over a 12 month period yielded mean values of 0.09 mg/L for fluoride ($F^-$), 2.12 mg/L for chloride ($Cl^-$), 0.03 mg/L for nitrite ($NO_2^-$), 1.07 mg/L for nitrate ($NO_3^-$), 0.32 mg/L for phosphate ($PO_4^{3-}$), 0.67 mg/L for sulphate ($SO_4^{2-}$), 0.45 mg/L for sodium ($Na^+$), 0.38 mg/L for ammonium ($NH_4^+$), 0.23 mg/L for magnesium ($Mg^{2+}$), 2.34 mg/L for potassium ($K^+$), and 1.94 mg/L for calcium ($Ca^{2+}$).

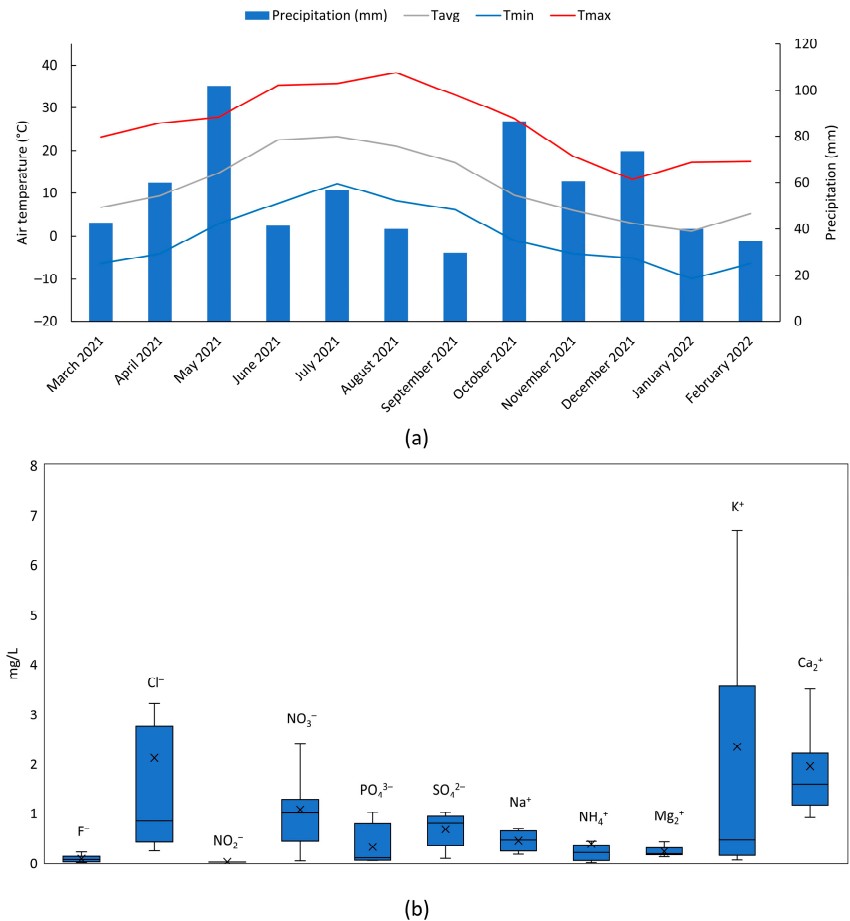

**Figure 2.** (**a**) Monthly precipitation and air temperature at the Velika Gorica meteorological station; and (**b**) chemical composition of precipitation in mg/L.

### 3.2. Characteristics of Soil Water

The descriptive statistics of soil moisture, soil temperature, isotopic, and chemical characteristics for soil water samples are listed in Table S1 (Supplementary Materials), while the graphical distributions in different soil horizons are shown in Figure 3. The table involves the mean, minimum, maximum, and standard deviation (SD) of each parameter. The results show that soil moisture through pedological profile ranges from 19.15 to 45.21% with the highest mean value in the A soil horizon (36.51%) and lowest in the C horizon (23.19%). The probe at the shallowest depth shows the greatest variability (SD is 9.32), ranging from 19.69 to 45.21%, while the deepest probe ranging from 19.15 to 24.79% shows the lower variability (SD is 2.15). Soil temperature ranges from 2.88 to 25.88 °C due to a seasonal variation with highest values in summer and lowest values in winter.

The values of $\delta^2$H and $\delta^{18}$O for soil water range from $-74.64$ to $-30.17$‰ and from $-10.68$ to $-4.24$‰, respectively. The value of SD decreased with depth, suggesting lower variability in isotopic composition. In Figure 4a it can be clearly seen that average values of isotopic composition from all soil horizons fall on the local meteoric water line (LMWL). However, the results also suggest that in the A and 2B soil horizons, the precipitation signature can be seen, while the 3BC and C soil horizons have a different isotopic signature. Furthermore, this can be also seen in Figure 4b, which confirms similar isotopic composition in the two deepest soil horizons in the observed time interval with almost no variation and response to precipitation.

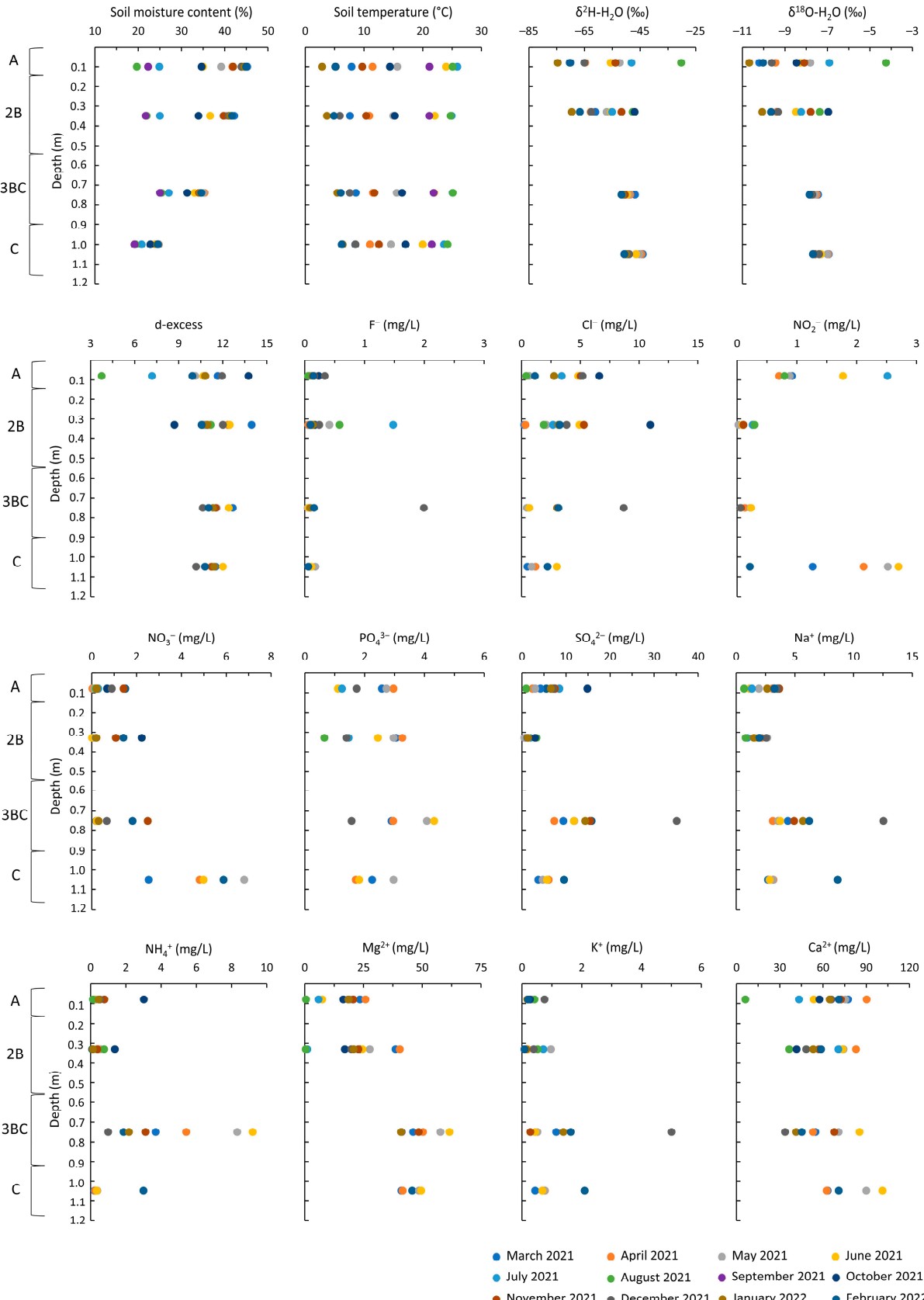

**Figure 3.** Distribution of soil moisture, soil temperature, isotopic, and chemical characteristics of soil water samples by depth.

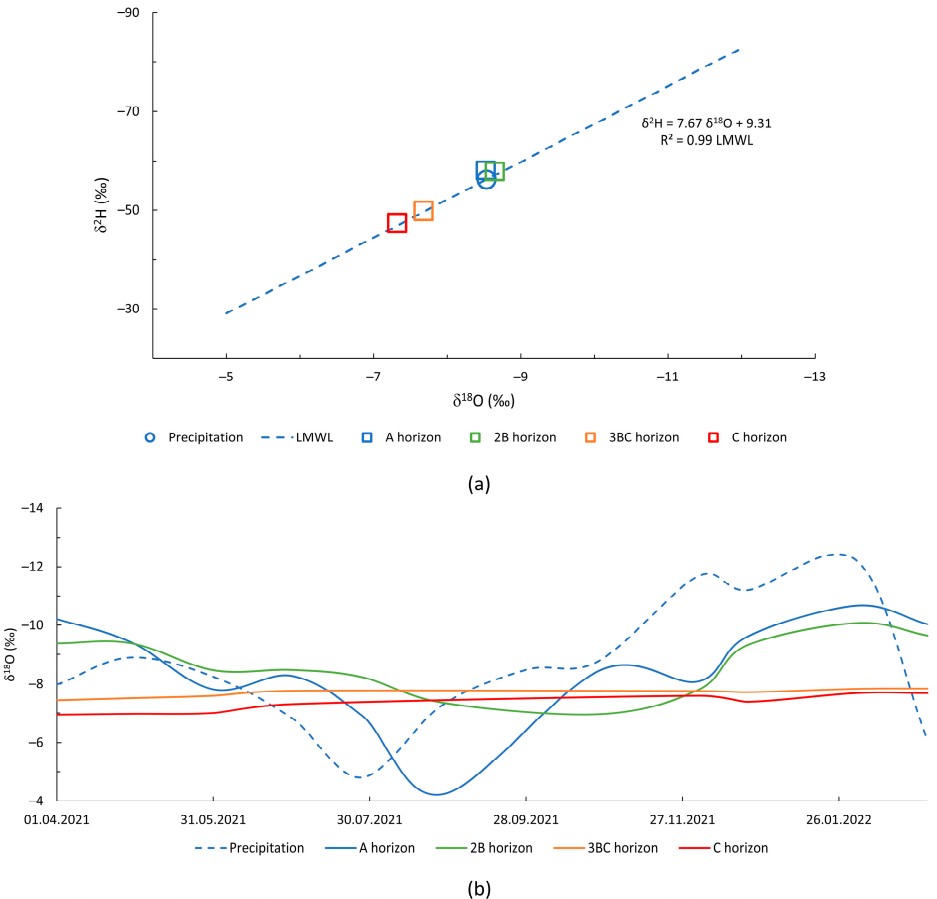

**Figure 4.** (**a**) Isotopic composition of soil water and precipitation and LMWL of Velika Gorica; and (**b**) variation in $\delta^{18}O$ in time in sampled soil water and precipitation.

D-excess ranges from 3.75 to 13.97‰ and shows smaller variability in the 3BC and C soil horizon (Table S1). Furthermore, smaller values of d-excess in the shallowest soil horizon in the summer months of 2021 (July and August) suggest influence of evaporation, which is consistent with the previous research where it was shown that in most cases evaporation fractionation is limited to the shallowest 0.3 m of soil [85], and which has also been observed in the A soil horizon within the previous research done at the study site [69].

Concerning the chemical composition of soil water, the $F^-$ and $Cl^-$ mean concentrations range from 0.11 to 0.32 mg/L and from 1.55 to 3.51 mg/L, respectively. The $Cl^-$ concentrations were higher in the colder months compared to the warmer months. A wide range of $NO_2^-$ and $NO_3^-$ concentrations were observed, with mean values range from 0.13 to 1.76 mg/L and from 0.52 to 5.00 mg/L, respectively. $NO_2^-$ content is higher in the A and C soil horizon, while $NO_3^-$ increases with depth. The nitrates stays at a low level in the first three soil horizons and then sharply rises in the C horizon (Figure 3). From Table S1, it can be observed that the SD of $NO_2^-$ and $NO_3^-$ at the deepest soil horizon is higher. The $PO_4^{3-}$ and $SO_4^{2-}$ mean concentrations range from 2.06 to 2.19 mg/L and from 1.50 to 15.15 mg/L, respectively. The $PO_4^{3-}$ and $SO_4^{2-}$ content in the 3BC soil horizon is higher than for the other depths. Further, a wide range of $NH_4^+$ concentrations were observed. The $NH_4^+$ mean concentrations range from 0.55 to 4.35 mg/L, with the higher content and wider range of value in the 3BC soil horizon. The $Na^+$ and $K^+$ mean concentrations range from 1.86 to 5.53 mg/L and from 0.31 to 1.36 mg/L, respectively. In comparison to other depths, the 3BC horizon exhibits higher $Na^+$ and $K^+$ concentrations. The $Mg^{2+}$ and $Ca^{2+}$ mean concentrations range from 16.15 to 48.47 mg/L and from 56.53 to 77.64 mg/L, respectively. $Mg^{2+}$ increases in the first three soil horizons and then decreases in the C horizon.

Figure 5 shows the variation of the $NO_3^-/Cl^-$ molar ratios in relation to $Cl^-$ concentrations. $NO_3^-/Cl^-$ molar ratios varied widely ranging from 0.03 to 0.75 with an average of 0.16 in the A soil horizon, from 0.002 to 0.33 with an average of 0.11 in the 2B horizon, from 0.04 to 0.47 with an average of 0.24 in the 3BC horizon and from 0.95 to 4.53 with an average of 2.46 in the C horizon. Therefore, compared to the first three soil horizons, the $NO_3^-/Cl^-$ molar ratios for C horizon had higher values.

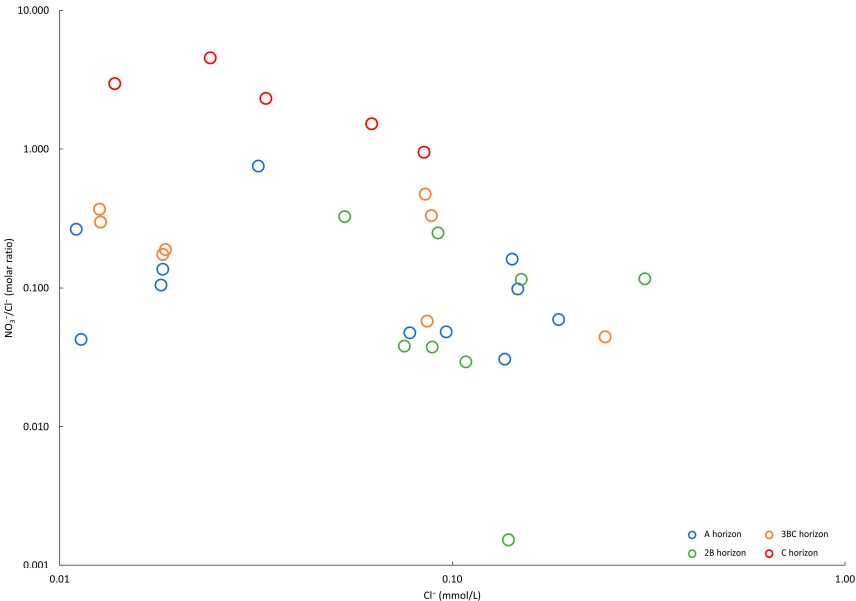

**Figure 5.** Relationship between $NO_3^-/Cl^-$ molar ratio and $Cl^-$ for A, 2B, 3BC, and C soil horizon.

In the investigated soil water samples within first soil horizon (A), the low $mNO_3^-/Cl^-$ ratio was accompanied by the high $mNO_2^-/Cl^-$ ratio (Figure 6). In the deepest soil horizon (C), under the relatively low soil moisture, the $mNO_3^-/Cl^-$ values are higher than $mNO_2^-/Cl^-$ values.

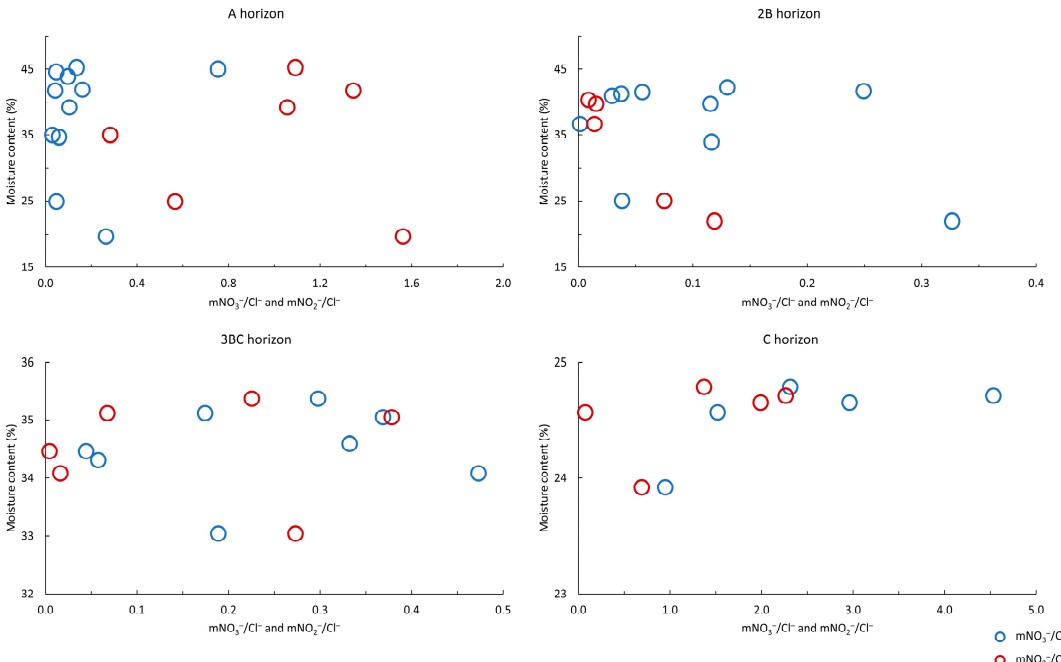

**Figure 6.** Relationship between soil moisture and $mNO_3^-/Cl^-$ or $NO_2^-/Cl^-$ for A, 2B, 3BC, and C soil horizon.

### 3.3. Analysis of Variance (ANOVA) and Kruskal-Wallis Test

For testing the assumptions of normality and homogeneity of variance, the Shapiro-Wilk (Table S2) and Levene's test (Table S3) are used, respectively. The test outcomes show that only four soil water parameters, i.e., temperature, $PO_4^{3-}$, $Mg^{2+}$ and $Ca^{2+}$, follow a normal distribution and have variance constant within each group. These variables were tested by one-way ANOVA. For the remaining 12 parameters, where the assumptions are not met, testing was conducted using the KW test.

The one-way ANOVA and KW test have been conducted to compare the variability of soil water parameters from a different sampling depth. The results of ANOVA (Table S4) indicate that there are no significant differences for soil temperature, $PO_4^{3-}$, and $Ca^{2+}$ within different sampling depth within the pedological pit. However, a statistically significant difference was observed for $Mg^{2+}$ between the different sampling depths. The results of KW test indicate that there are significant differences for soil moisture, $\delta^2H$, $\delta^{18}O$, $NO_2^-$, $NO_3^-$, $SO_4^{2-}$, $Na^+$, $NH_4^+$, and $K^+$. Conversely, d-excess, $F^-$ and $Cl^-$ are an insignificant difference within the four sampling groups (Table S5).

Tukey HSD test is utilized after one-way ANOVA only for the $Mg^{2+}$ parameter (Table S6), where a significant difference is observed. The *p*-values of $Mg^{2+}$ between A and 2B, as well as between 3BC and C, indicate an insignificant difference. Among all other groups the *p*-values indicate the significant difference. Mann–Whitney U comparison test (Table S7) is used after the KW test for nine parameters with a significant difference. The results indicate that the different soil horizon groups of moisture content (between A and C, 2B and C, 3BC and C), $\delta^2H$ (between A and 3BC, A and C, 2B and 3BC, 2B and C), $\delta^{18}O$ (between A and 3BC, A and C, 2B and 3BC, 2B and C, 3BC and C), $NO_2^-$ (between A and 2B, A and 3BC, 2B and C, 3BC and C), $NO_3^-$ (between A and C, 2B and C, 3BC and C), $SO_4^{2-}$ (between A and 2B, A and 3BC, 2B and 3BC, 2B and C, 3BC and C), $Na^+$ (between A and 3BC, 2B and 3BC, 2B and C), $NH_4^+$ (between A and 3BC, 2B and 3BC, 3BC and C) and $K^+$ (between A and 3BC, A and C, 2B and 3BC, 2B and C) are remarkably different. There is no statistically significant difference observed among all the other independent groups.

### 3.4. Characteristics of Soil

The distribution of 16 soil parameters through depth is presented in Figure 7. As shown, soil pH values tend to increase with depth. Soil pH is usually below 7, ranging from 6.4 to 6.9, except for in the deepest interval (C soil horizon) where pH of 7.1 was measured. The EC values range from 80.5 to 150.6 μS/cm. TC and TIC content decreases with depth and then increases sharply at the bottom of the profile. TOC decreases with depth and ranges from 0.4 to 2.3%. Contents of $TiO_2$, $Fe_2O_3$ and MnO range from 0.3 to 0.9 wt.%, from 5.7 to 8.5 wt.% and from 0.1 to 0.2 wt.%, respectively. In comparison to other depths, the C soil horizon demonstrates the lowest $TiO_2$, $Fe_2O_3$, and MnO content. Cr ranges from 316 to 415 mg/kg and changes dramatically along the depth, with all measured values exceeding the maximum permissible limits for soil. Ni, Zn, Pb, and As concentrations range from 54 to 93 mg/kg, from 92.4 to 152.6 mg/kg, from 27.8 to 45.4 and from 13.4 to 23.6 mg/kg, respectively. Ni content in the 2B soil horizon exceeds the maximum permissible limits for soil. Ni, Zn, Pb, and As concentrations are the lowest in C soil horizon. A wide range of Co concentrations, from 35 to 89 mg/kg, was observed through pedological profile. Hg ranges from 7.8 to 11 mg/kg and all measured values exceed the maximum permissible limits. Clay content varies from 9.5 to 20.8%, and sand content varies from 0.2 to 6.3%, with the highest quantities observed in the C soil horizon. Silt content ranges from 72.9 to 89.5% and generally decreases with depth. Based on the FAO [75] soil texture classification, soil samples up to 60 cm deep are classified as silt and soil samples from 0.60 to 1.2 m are classified as silt loam, with one interval (0.90–1.0 m) being classified as silt.

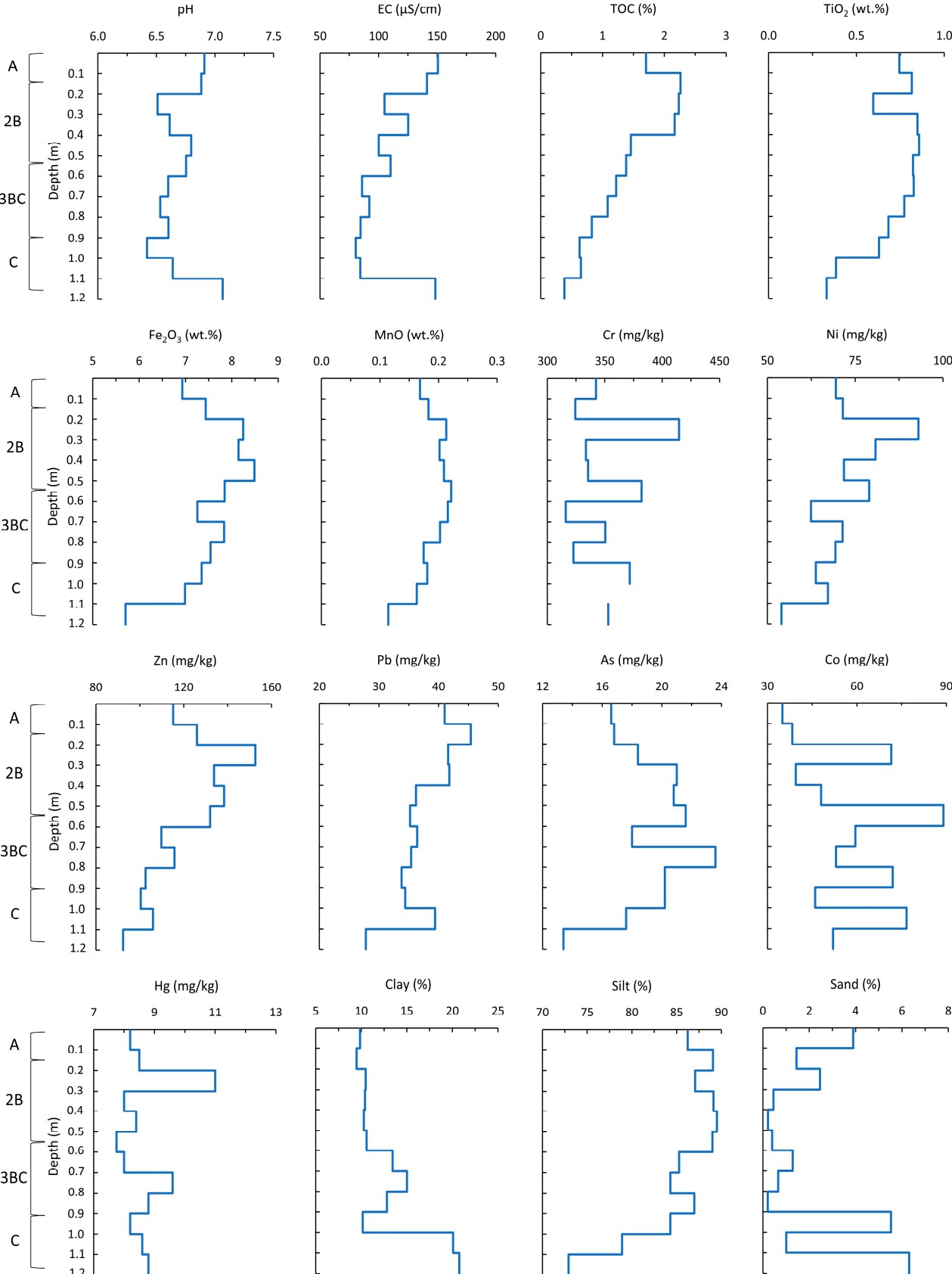

**Figure 7.** Distribution of pH, EC, TOC, TiO$_2$, Fe$_2$O$_3$, MnO, heavy metals, and soil texture of soil samples by depth.

## 4. Discussion

### 4.1. Nitrogen Species Distribution within the Soil Zone

Although the accumulation of $NH_4^+$ in soils is not common [3], $NH_4^+$ is the dominant N species in the 3BC soil horizon in the investigated soil profile. High concentrations of $NO_2^-$ were detected in A and C soil horizon. In the first horizon, the rate of nitritation is faster than nitratation, resulting in the accumulation of $NO_2^-$ [23,24]. $NO_3^-$ increases with depth and, together with $NO_2^-$, represents the dominant species of N in the deepest soil horizon (Figure 3). Additionally, results suggest that N species distribution along soil profile depends on various factors.

Soil water content generally shows lower variability at greater soil depths [69,86], which is in line with the findings of this study. Given that soil tends to become dry and lose moisture during rainless and dry periods [87], the lowest soil moisture was observed for July, August, and September because those months were periods with precipitation below monthly average precipitation for this area. The highest concentration of $NO_3^-$ in the C soil horizon, was observed in May, coinciding with a sudden increase in rainfall (Figure 2a) that led to enhancement of $NO_3^-$ leaching [9]. Conversely, the lower $NO_3^-$ concentrations observed in the first three soil horizons are a result of elevated water content, which in turn leads to oxygen limitation and reduced nitrification rate [9,28]. In contrast, the degree of denitrification in the deepest soil horizon is constrained by lower soil moisture levels, causing an increase in $NO_3^-$ concentration [29]. Soil moisture through pedological profile has the highest mean value in the A horizon and lowest in the C horizon (Table S1). After the third soil horizon (3BC), soil moisture drops, likely due to reduced retention resulting from the presence of coarse-grained particles (Table 1).

According to Ayiti and Babalola [9], $NO_2^-$ concentrations are higher during warmer months (June and July) when the temperature and microbial activity are higher, while there are no recorded concentrations higher than 0.2 mg/L during the coldest months (December, January, and February). $NO_3^-$ concentrations are reduced during warmer months, particularly in the first three soil horizons, likely due to assimilation of $NO_3^-$ through plants and denitrification process [11]. Considering that the pedological profile is situated in an area without potential anthropogenic sources of nitrate and that C soil horizon probably cannot retain the most of precipitation that infiltrates [69], which corresponds to different isotopic signature of precipitation and absence of variation in isotopic composition of soil water in C soil horizon within this research (Figure 4b), the examination of variation of the $NO_3^-/Cl^-$ molar ratios in relation to $Cl^-$ concentrations could be critical for the definition of nitrates produced by soil N nitrification. As shown in Figure 5, the highest values of molar ratios are observed in the C soil horizon with an average of 2.5 and minimum of 0.9 while the highest molar ratio in all other soil horizons is 0.8. This corresponds to research that has shown that higher $NO_3^-/Cl^-$ molar ratios relative to $Cl^-$ concentrations can suggest the occurrence of $NO_3^-$ concentrations, which are the consequence of nitrification of soil N [2].

This corresponds to the evaluation of molar relationship of $NO_3^-$, $NO_2^-$, and $Cl^-$ and soil water content. The observed low $mNO_3^-/Cl^-$ ratio and high $mNO_2^-/Cl^-$ ratio within the first soil horizon (A) could likely be attributed to an oxygen-deficient environment [61]. Under that condition, both nitrification and denitrification are relatively strong and $NO_2^-$ is the intermediate product, resulting in a large amount $NO_2^-$ accumulation. Within the C soil horizon, the $mNO_3^-/Cl^-$ values are higher than $mNO_2^-/Cl^-$ (Figure 6), under the relatively low soil moisture levels and higher oxygen content, leading to stronger nitrification.

### 4.2. Variability of Soil Water Parameters within the Soil Zone

The statistical results indicate that there are significant differences for soil moisture, $\delta^2H$, $\delta^{18}O$, $NO_2^-$, $NO_3^-$, $SO_4^{2-}$, $Na^+$, $NH_4^+$, $Mg^{2+}$, and $K^+$, but show no significant differences across various sampling depths within the pedological profile for soil temperature, d-excess, $F^-$, $Cl^-$, $PO_4^{3-}$, and $Ca^{2+}$ (Tables S4 and S5). $Mg^{2+}$ across the first two, as well as along the third and fourth soil horizon, has a similar distribution. Soil moisture and

$NO_3^-$ have identical distribution patterns across the first three soil horizons. The deepest soil horizon (C) stands out as the sole horizon with a distinct distribution for these two parameters. Deuterium, $Na^+$, and $K^+$ have a similar distribution within the first two and the last two soil horizons. Oxygen-18 has the same distribution pattern only within the first two soil horizons. $NO_2^-$ across second and third, as well as within first and fourth horizons, has identical distribution. The only soil horizon within $NH_4^+$ that has a distinct distribution is the third one (3BC). The soil horizons with the same $SO_4^{2-}$ distribution are the first and fourth.

### 4.3. Factors Influencing $NO_3^-$ Distribution

Vertical distribution of $NO_3^-$ content in the soil profile is influenced by pH [34], soil moisture [28], and organic carbon availability [20,38]. The distribution of soil pH and EC appears to be quite similar throughout the soil profile. Soil pH values increase with the soil depth, while on the other hand, organic carbon decreases with the soil depth [40]. This occurs mainly due to the higher OM in the topsoil (Table 1), which possibly leads to a pH reduction through the decomposition of OM [35,56]. In the first three soil horizons with lower pH values, the intensity of nitrification is reduced compared to the deepest soil horizon with a higher soil pH value [32–34]. Similar to recent research [9], the deepest soil horizon (C) with a pH above 7 has a prerequisite for optimum activity of ammonia and nitrite oxidizers, i.e., for nitrification. The $NO_3^-$ is low in the first three soil horizons and then increases in C horizon, likely because of a decrease in the TOC content, as low organic carbon levels may increase the amount of $NO_3^-$ [20,32,38]. The A horizon contains the highest value of OM (Table 1) which should contribute to faster water percolation and permeability [39]. Additionally, drastic reduction in soil water content can also have significant impact on the denitrification rates [88]. The same research also showed that soil nitrification can both decrease and increase depending on the soil water content. Soil nitrification increased with an increase in soil moisture when soil water content was less than approximately 27% and decreased with an increase in soil moisture if it was above 27%. Considering that clay content and the specific surface area of the soil are associated with hysteresis caused by the adsorbed water content in the soil [89], the soil horizon with the insignificant amount of clay, i.e., C soil horizon (Table 1), is characterized with the lowest soil moisture. It must be emphasized that maximum water content in C soil horizon did not exceed 25%, which also suggests more dominant occurrence of soil nitrification. Si and Kachanoski [90] and Zhang et al. [91] have shown that hysteresis effects can influence water transfer, microbial activities, as well as solute transport in soil. From that perspective it is important to investigate how and if nitrogen-related processes depend on the hysteresis effect. It was shown that the hysteresis effect can be different in multiple cycles of drying and wetting [92]. Furthermore, influence of soil shrinkage should also be investigated in future research because it is known that nitrogen-related processes depend on oxygen availability. Some research has shown that void ratio after soil shrinkage can have considerable influence on soil water characteristic curve [87]. These results suggest that both hysteresis and soil shrinkage can influence oxygen concentrations in soil, which can directly affect nitrogen transformation and the related processes. According to Six et al. [55], TOC contents are higher at sampling intervals where soil is rich in silt and poor in sand (2B and 3BC soil horizons), most likely due to silt particles which stabilize soil OM from being decomposed by microorganism. Conversely, reduced input of OM along C soil horizon, characterized by a higher presence of sand, tends to decrease TOC content. The presence of $TiO_2$, $Fe_2O_3$, and MnO [41,42,45] up to a depth of 0.9 m (Figure 7) could represent another important factor contributing to the reduction in nitrification rates. Almost all heavy metal concentrations exhibit lower concentrations in C horizon and, as mentioned, often negatively affect the nitrification rate and inhibit the activity of microorganisms [46,47], which can lead to an accumulation of $NH_4^+$ and reduction of $NO_3^-$ in the soil within the first three soil horizons. Since accumulation of $NH_4^+$ in soils is typically uncommon [3], its presence at a certain depth indicates the existence of reducing

conditions [8]. The highest accumulation of $NH_4^+$ is observed in the 3BC soil horizon, characterized by a notable presence of silt and clay, which also points to the existence of oxygen depleted conditions. Under such conditions, possibly both Fe and Mn act as electron acceptors resulting in the production of $NO_3^-$ [44] and $NO_2^-$ [43] as the end products in C soil horizon. Additionally, as noted by Varnier et al. [8], a significant amount of $NH_4^+$ in 3BC soil horizon could be evidence of incomplete nitrification. According to previous research, starting from a depth of 0.9 m (i.e., the C horizon), which is characterized by a significant presence of gravel, aerobic conditions may prevail and $NO_3^-$ may accumulate.

## 5. Conclusions

This study used the characteristics of soil water and soil to evaluate the distribution and migration of N compounds at different soil profile depths. Statistical techniques were used to explore the significant variability of soil water parameters from different depths. It has been shown that soil moisture content, pH, TOC, and soil texture are important factors influencing the concentrations of N species within the soil zone. In addition, results reveal that all N species are present in soil water from all soil horizons. Moreover, the shallowest soil horizon has the highest $NO_2^-$ concentrations, which suggest the dominance of nitritation. On the other hand, $NH_4^+$ is dominant in 3BC soil horizon, which indicates the oxygen-deficient environment of the soil zone, while in the deepest C soil horizon N is present mostly as $NO_3^-$, which suggests the dominance of nitrification under aerobic conditions. Considering that nitrates are very soluble and have leaching potential through soil zone, the aerobic conditions of the C soil horizon promote stability and the potential for $NO_3^-$ transport. Additionally, it has been shown that $NO_3^-$ concentrations are a result of nitrification of soil N, which is confirmed by the absence of precipitation isotopic signature and higher $NO_3^-/Cl^-$ molar ratios when observing all soil horizons. Results also suggest that if $NO_3^-$ are produced from the nitrification of the soil N within the Eutric Cambisols it should have a $NO_3^-/Cl^-$ molar ratio higher than 1. Within the coarse-grained Eutric Cambisols, where anthropogenic sources of nitrate are not present, results suggest that soil nitrification of $NH_4^+$ is more common, while N primarily migrates to deeper parts of unsaturated zone in the form of $NO_3^-$ and $NO_2^-$. The long-term goals include modelling of flow and N compounds transport within the soil and unsaturated zone, which is expected to enhance our comprehension of the entire aquifer system and facilitate the sustainable management of the Zagreb aquifer.

**Supplementary Materials:** The following supporting information can be downloaded at: https://www.mdpi.com/article/10.3390/su152316529/s1, Table S1: Descriptive statistics of soil moisture content, soil temperature, isotopic, and chemical compositions for soil water samples; Table S2: The results of testing the assumptions of normality of soil moisture content, soil temperature, isotopic, and chemical compositions for soil water samples; Table S3: The results of testing the assumptions of homogeneity of soil moisture content, soil temperature, isotopic, and chemical compositions for soil water samples; Table S4: The result of the ANOVA test of soil temperature, $PO_4^{3-}$, $Mg^{2+}$, and $Ca^{2+}$ within the four groups; Table S5: The result of the Kruskal–Wallis test of soil moisture, $\delta^2H$, $\delta^{18}O$, d-excess, $F^-$, $Cl^-$, $NO_2^-$, $NO_3^-$, $SO_4^{2-}$, $Na^+$, $NH_4^+$, and $K^+$ within the four groups; Table S6: The result of Tukey HSD post hoc comparison between groups; Table S7: The result of the Mann–Whitney U post hoc comparison between groups.

**Author Contributions:** Conceptualization, P.B.; formal analysis, P.B.; investigation, P.B.; methodology, P.B.; writing—original draft, P.B.; visualization, P.B. and J.P.; supervision, J.P. and Z.K.; writing—review and editing, J.P., Z.K., T.B. and M.P. All authors have read and agreed to the published version of the manuscript.

**Funding:** This research received no external funding.

**Institutional Review Board Statement:** Not applicable.

**Informed Consent Statement:** Not applicable.

**Data Availability Statement:** Data will be made available on request.

**Acknowledgments:** The work was part of the Young Researches' Career Development Project—Training New Doctoral Students (DOK-2020-01) supported by the Croatian Science Foundation (HRZZ) and IAEA TC project CRO7002 "The use of nitrogen and oxygen stable isotopes in the determination of nitrate origin in the unsaturated and saturated zone of the Velika Gorica well field". The authors would like to thank Laura Bačani for providing data from the research polygon. Thanks also go to Saša Šipek and Ivan-Borna Balaž for taking soil samples.

**Conflicts of Interest:** The authors declare no conflict of interest.

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
