# Peer review of "Determination of Nitrate Migration and Distribution through Eutric Cambisols in an Area without Anthropogenic Sources of Nitrate (Velika Gorica Well Field, Croatia)"

_sustainability, doi:10.3390/su152316529_

Round 1

Reviewer 1 Report

Comments and Suggestions for Authors

The manuscript titled ‘’ Determination of nitrate migration and distribution through Eutric Cambisols in an area without anthropogenic sources of nitrate (Velika Gorica well field, Croatia)’’ is very interesting work with a strong structure! Honestly, I really enjoyed reading the text as it was written very precise, well-expended, and used enough references. Results were performed professionally in an aesthetic manner! And the discussion came over the interpretation of the findings well. However, there is one point in my view. The authors did not refer to any practical solution for minimizing nitrate leaching! Lines 28-29 refer to the adverse of some commercial organic sources which may affect the nutrient status in soil. But, on the other hand, there ARE some beneficial soil amendments such as zeolite, biochar, … which definitely can decrease nutrient leaching without any adverse. These amendments were broadly investigated over recent years. So it is better to distinguish between sources! Moreover, it helps to avoid a bit of bias that could be implied from the current form. As a suggestion please add 3-4 lines related to this issue. Here is one ref you can use: https://doi.org/10.1007/s11270-022-05910-4

Good luck!

Reviewer 2 Report

Comments and Suggestions for Authors

Comments to the Author

The study's objectives encompass the assessment of nitrate (NO3-) distribution and movement within a pedological profile situated in an area devoid of anthropogenic nitrate sources. These objectives have been addressed through the application of appropriate statistical methods aimed at discerning noteworthy variations across distinct soil horizons. Furthermore, the study delves into the analysis and characterization of the factors that impact the concentration of NO3- within the soil profile. The paper is interesting. However, some comments need to be solved. It is recommended that major revisions to address these concern.

1. The “2.2 Field and laboratory measurements”section you provided is detailed and comprehensive. It offers a clear understanding of the data collection and analysis process. However, I would like to make a few suggestions to enhance the clarity and completeness of this section: Consider breaking down this section into subheadings for easier readability. Subheadings such as "Meteorological Data Collection," "Soil Sampling and Analysis," and "Laboratory Measurements" can help readers navigate through the information more efficiently.

2. Soil Sampling: Provide more information on the sampling methodology. Describe the number and distribution of soil sampling points, especially if it has relevance to the study's objectives. Additionally, explain the reasons behind not collecting soil water samples in certain months and how this may impact the study's findings.

3. By addressing these points, the methodology section will become more informative and transparent, allowing readers to better understand the data collection and analysis processes.

4. How do the findings of Ayiti and Babalola [15] or Six et al. [53] align or differ from your observations? More literature should be considered  Journal of Hydrology, 2023, 618, 129230; Journal of Hydrology, 2022,614, 128583; Engineering Geology, 2021,284, 106035; Soil Science Society of America Journal, 2020, 84(5), 1642-1649; Acta Geotech. 2020,15:3321-3326

5. The discussion should be restructured to improve clarity and flow. Consider organizing the discussion into subsections that correspond to key findings, factors influencing nitrogen distribution, and implications. This will help readers navigate through the information more effectively and understand the relationships between different variables.

6. In-Depth Interpretation: Provide a more in-depth interpretation of the observed patterns, particularly regarding the accumulation of NO3- in the C soil horizon and the implications of the NO3-/Cl- molar ratios. Make sure to explicitly link your findings with the study's objectives and offer a comprehensive discussion of the ecological significance of the results.

Reviewer 3 Report

Comments and Suggestions for Authors

Dear Authors and Editors,

The manuscript "Determination of nitrate migration and distribution through Eutric Cambisols in an area without anthropogenic sources of nitrate (Velika Gorica well field, Croatia)" contains interesting and valuable information. The water quality of the well fields is of primary importance. The authors examined many parameters for the study, which is a great advantage of the manuscript. At the same time, it is a pity that for the soil analysis, a soil sample was taken from only one point, sampling every 10 cm to a depth of 1.2 m (line 182-187). I couldn't find any information on how far apart the soil sampling and groundwater sampling took place.

Other comments:

It is unnecessary to write about future objectives in the Abstract, but the mention of the Zagreb aquifer is important (line 19-21).

There are some unclear details between lines 163-181. It is not clear on what basis the mentioned 38 soil water samples were selected (4 layers x (12 – 1) months = 44 samples?). Please clarify the number of soil water samples. Also, the depth of the soil water samples should be indicated in this paragraph.

Reference [81] in line 243 is not justified.

The description of the statistical method is a bit exaggerated. Emphasizing the "suitable statistical technique" mentioned in several places is unnecessary, because it is mostly natural in scientific papers. I feel that Figure 1 is unnecessary, since it is also described in text. It is also unnecessary to emphasize in Section 3.3 what is greater than or below the significance level of 0.05. It is enough to declare what is significantly different and what is not.

In Supplementary Tables 6 and 7, instead of a pairwise comparison, it would be more transparent to indicate only the mean values of the parameters, and to mark significantly different means with different letters.

The abbreviation LMWL in line 280 should be explained at the first occurrence or the whole expression should be used.

The concept and significance of d-excess should be explained in the Introduction.

Based on the above, I recommend a minor revision of the manuscript.

Round 2

Reviewer 2 Report

Comments and Suggestions for Authors

Accept